# Prior Influenza Infection Mitigates SARS-CoV-2 Disease in Syrian Hamsters

**DOI:** 10.3390/v16020246

**Published:** 2024-02-03

**Authors:** Caterina Di Pietro, Ann M. Haberman, Brett D. Lindenbach, Peter C. Smith, Emanuela M. Bruscia, Heather G. Allore, Brent Vander Wyk, Antariksh Tyagi, Caroline J. Zeiss

**Affiliations:** 1Department of Pediatrics, Yale School of Medicine, New Haven, CT 06519, USA; caterina.dipietro@yale.edu (C.D.P.); emanuela.bruscia@yale.edu (E.M.B.); 2Department of Immunobiology, Yale School of Medicine, New Haven, CT 06519, USA; ann.haberman@yale.edu; 3Department of Microbial Pathogenesis, Yale School of Medicine, New Haven, CT 06519, USA; brett.lindenbach@yale.edu; 4Department of Comparative Medicine, Yale School of Medicine, New Haven, CT 06519, USA; peter.smith@yale.edu; 5Department of Internal Medicine, Yale School of Medicine, New Haven, CT 06519, USA; heather.allore@yale.edu (H.G.A.); brent.vanderwyk@yale.edu (B.V.W.); 6Department of Biostatistics, Yale School of Public Health, New Haven, CT 06519, USA; 7Department of Genetics, Yale Center for Genome Analysis, New Haven, CT 06519, USA; antariksh.tyagi@yale.edu

**Keywords:** SARS-CoV-2, influenza, hamster, interference, transcriptomics

## Abstract

Seasonal infection rates of individual viruses are influenced by synergistic or inhibitory interactions between coincident viruses. Endemic patterns of SARS-CoV-2 and influenza infection overlap seasonally in the Northern hemisphere and may be similarly influenced. We explored the immunopathologic basis of SARS-CoV-2 and influenza A (H1N1pdm09) interactions in Syrian hamsters. H1N1 given 48 h prior to SARS-CoV-2 profoundly mitigated weight loss and lung pathology compared to SARS-CoV-2 infection alone. This was accompanied by the normalization of granulocyte dynamics and accelerated antigen-presenting populations in bronchoalveolar lavage and blood. Using nasal transcriptomics, we identified a rapid upregulation of innate and antiviral pathways induced by H1N1 by the time of SARS-CoV-2 inoculation in 48 h dual-infected animals. The animals that were infected with both viruses also showed a notable and temporary downregulation of mitochondrial and viral replication pathways. Quantitative RT-PCR confirmed a decrease in the SARS-CoV-2 viral load and lower cytokine levels in the lungs of animals infected with both viruses throughout the course of the disease. Our data confirm that H1N1 infection induces rapid and transient gene expression that is associated with the mitigation of SARS-CoV-2 pulmonary disease. These protective responses are likely to begin in the upper respiratory tract shortly after infection. On a population level, interaction between these two viruses may influence their relative seasonal infection rates.

## 1. Introduction

The projected endemic pattern of SARS-CoV-2 in the Northern hemisphere is likely to overlap temporally with that caused by seasonal influenza and other respiratory viruses, with infections peaking in the cold season [1,2,3]. Influenza and SARS-CoV-2 coinfections have attracted particular attention [4,5,6] because both viruses can cause severe pulmonary or multi-organ disease [7,8]. Consequently, the potential synergism accompanying coinfection is of concern. This outcome is supported by reports of more severe disease in patients with influenza and SARS-CoV-2 coinfection [4,5,9,10,11]. Enhanced disease severity resulting from simultaneous influenza/SARS-CoV-2 coinfection has been demonstrated in ferrets [12], hamsters [13,14] and mice [15].

However, several lines of evidence implicate more nuanced interactions during viral coinfection. Some influenza/SARS-CoV-2 coinfected patients experience less severe disease [6] or disease severity similar to that induced by SARS-CoV-2 infection alone [16,17]. At a population level, epidemiologic studies indicate that respiratory viral interactions may be synergistic or competitive in ways that influence the prevalence of individual infections [18,19]. Competitive interactions between respiratory viruses may also influence disease severity [20]. For example, rhinovirus and respiratory syncytial virus coinfection appear to attenuate the severity of influenza in children [21]. In animals, immune priming of the respiratory tract by prior infection with one respiratory virus reduces morbidity in subsequent infection with an unrelated respiratory virus [22,23,24]. In vitro studies confirm that prior infection with rhinovirus can limit subsequent SARS-CoV-2 replication [25,26,27]. Similar protection is achieved by intranasal administration of live attenuated influenza vaccine in SARS-CoV-2-infected ferrets [28].

Therefore, it appears that the introduction of SARS-CoV-2 into the respiratory virus landscape could result in a range of outcomes. As shown by in vitro data [25,26], the timing of coinfection is critical. In this study, we established that H1N1 given 48 h prior to SARS-CoV-2 largely protected animals from disease and that some aspects of this protection may become operational within hours of H1N1 infection.

## 2. Materials and Methods

All animal work was conducted in a Biosafety Level 3 (BSL3) facility under an approved Yale Institutional Animal Care and Use Committee protocol (2020-20342). Yale University is registered as a research facility with the United States Department of Agriculture and is accredited by AAALAC “https://www.aaalac.org (accessed on 1 February 2024)”.

### 2.1. Viruses and Propagation

SARS-CoV-2 Wuhan/IVDC-HB-01/2019 isolate was launched from cloned cDNA [29] by RNA transcription with T7 RNA polymerase and RNA transfection into baby hamster kidney (BHK) cells engineered to express the SARS-CoV-2 N gene and mixed with Vero E6 cells. Electroporation conditions were as previously described [30]. The primary SARS-CoV-2 stock was passaged once in Vero E6 cells, yielding a titer of 2.5 × 10^7^ PFU/mL). Upon sequencing of the virus stock, a single-nucleotide alteration (Arg685Ser) was identified in the Furin cleavage site of the Spike gene, most likely due to propagation in Vero E6 cells [31]. Influenza virus A/Michigan/45/2015 (H1N1) pdm09 (hereafter referred to as H1N1 in the text) was obtained from the International Reagent Resource and propagated once in MDCK-London cells, yielding a titer of 2.18 × 10^7^ PFU/mL. Viruses were titered by plaque assay with Dulbecco’s modified Eagle medium containing 2% fetal calf serum and 0.6% Avicel CL-611 (FMC Biopolymers), fixation with 7% formaldehyde and staining with 1% (*w*/*v*) crystal violet in 20% (*v*/*v*) ethanol.

### 2.2. Animals and Housing

Eight-week-old female and male Syrian Golden hamsters (*Mesocricetus auratus*) weighing between 90 and 120 g were purchased from Envigo (Indianapolis, IN, USA). Animals were individually housed in filter top cages on corncob bedding with cotton neslets and provided ad lib access to autoclaved pellets (2018S, Envigo, Somerset, NJ, USA) and chlorinated water. Rooms were maintained at 72 °F on an evenly split light cycle (7 AM/7 PM). Hamsters were acclimated for 5–7 days prior to infection.

### 2.3. Anesthesia, Viral Inoculation, Clinical Evaluation and Euthanasia

Hamsters were anesthetized briefly by using the open drop method (isoflurane: propylene glycol 30% *v*/*v*). Intranasal inoculation was performed with 1 × 10^6^ plaque-forming units (PFU) per animal of SARS-CoV-2 in DMEM or H1N1 in OptiMEM II. A total volume of 48 µL (SARS-CoV-2) or 54 µL (H1N1) was delivered split evenly between nostrils. Animals were fully recovered within 1–2 min of inoculation. Hamsters were weighed once daily between 8 and 10 AM, following cage-side collection of respiratory rate when animals were still asleep. Animals were euthanized by using 100% isoflurane via the open drop method, followed by creation of pneumothorax. For experiment 1, animals were given H1N1 first followed by SARS-CoV-2 given either 3 h (referred to as 3 h coinfected group) or 48 h (referred to as 48 h coinfected group) later. Control animals were inoculated with similar volumes of DMEM or OptiMEMII media only. Twenty-four animals split evenly by sex were used per infection group, thus providing 6 hamsters (3 males and 3 females) at each sacrifice time point (2, 4, 7 and 10 days post-infection (dpi); Figure 1). A similarly sex-balanced 24-animal control cohort was aggregated from 4 to 6 animals accompanying each infection group. For experiment 2, six animals were used per infection group (H1N1 alone, SARS-CoV-2 alone and H1N1 given 48 h prior to SARS-CoV-2) using similar virus isolates, doses and volumes as used in the prior experiment. Four media-only inoculated animals provided controls. Half of the animals in each infection group were sacrificed 24 h after all infections, and the remainder sacrificed at 48 h. All animals in this experiment were male.

### 2.4. Pathology, Immunohistochemistry and Microscopy

Following bronchoalveolar lavage, the right bronchus was ligated with 3.0 silk sutures and the left lung intratracheally infused with 5 mL/kg (half the total lung tidal volume) of 4% paraformaldehyde (PFA): 0.25% low-molecular-weight agarose, followed by ligation of the trachea, removal of the remaining lung and immersion in 4% paraformaldehyde for 48 h. Remaining tissues (brain, skull, heart, tongue, esophagus, stomach, large and small intestines, liver, spleen, kidney, reproductive tract, pancreas, cervical and mesenteric lymph nodes, salivary glands) were fixed in 10% neutral-buffered formalin. Following gross photography, left lungs were cut sagitally and the entire lung cassetted. Lungs and other soft tissues were submitted for standard formalin-fixed paraffin-embedding (FFPE), processing, and sectioning at 5 µm. Lung histopathology was assessed in hematoxylin and eosin (H&E)-stained sections using a semi-quantitative scoring system adapted from published criteria [32] that included airway, vascular and parenchymal components to arrive at a total histopathology score (Appendix A). Scoring was performed by a board-certified veterinary pathologist blinded to slide identity. Nasal pathology was performed after immersion of the head in 4% paraformaldehyde for 1 week, followed by decalcification for 24 h, coronal sectioning of nasal passages at three levels [33], FFPE and sectioning at 5 µm. Immunohistochemistry was performed on deparaffinized rehydrated slides (Appendix A) using standard methods [34] following antigen retrieval with 10 mM sodium citrate (pH 6.0). Light microscopic images were taken using a Zeiss Axioskop and with Axiocam MrC camera. Confocal images were taken with a Zeiss laser scanning microscope 880 with Airyscan, and images processed using Zeiss Zen Blue 3.1 software.

### 2.5. Flow Cytometry (Blood and Bronchoalveolar Lavage Fluid)

Animals were bled by cardiac puncture following euthanasia and blood collected in heparinized tubes. An 18 G venous catheter was inserted into the exposed trachea and secured with 4.0 silk ligatures. Bronchoalveolar lavage (BAL) was performed with 2 mL of 1× phosphate-buffered saline 0.1 mM EDTA with protease inhibitor (Roche cat # 11873580001). Blood and BAL samples were centrifuged at 400 rcf, supernatants removed for other analyses and the pellets resuspended in ACK. Following ACK treatment, samples were resuspended in 4% PFA and incubated at RT for 30–45 min, centrifuged and resuspended in staining buffer (1× phosphate-buffered saline + 3% Fetal Bovine Serum) prior to removal from the BSL3 facility. To coordinate the staining and data collection of multiple time points across cohorts, all samples were cryopreserved in freezing media (90% FBS 10% DMSO) and thawed later for concurrent antibody staining. Samples were centrifuged and the pellets first resuspended in 50 µL of a blocking buffer (20% FBS, 2% rat serum and 2% mouse serum in PBS) for 30 min on ice. An antibody cocktail (anti-I-Ek, anti-immunoglobulin, anti-Aif1 as listed in Appendix A) in staining buffer was then directly added in a 50 µL volume, further incubated 1 h on ice and washed twice. Samples were stained with secondary reagents, further washed and resuspended in a DNA binding dye (Vybrant Violet Ready for Flow) to assist in distinguishing intact nucleated cells from debris and RBCs. Data were acquired on a BD LSRII flow cytometer and further analyzed by using FlowJo v10 software (Treestar) using the gating strategy shown in Appendix A.

### 2.6. Nasal Transcriptomics

Using sterile instruments, we adapted a published protocol [35] to remove nasal bones, expose underlying turbinates and manually strip nasal epithelia followed by immediate submersion in 0.5 mL Trizol (Waltham, MA, USA). Following tissue homogenization (gentleMACS™ Dissociators, Miltenyi Biotec, Bergisch Gladbach, Germany), RNA was extracted (RNeasy Mini Kit; Qiagen, Germantown, MD, USA) and potential DNA contamination was removed with 30 min incubation in RNase-free DNase (Qiagen). Total RNA quality was determined by estimating the A260/A280 and A260/A230 ratios by nanodrop. RNA integrity was determined by running an Agilent Bioanalyzer gel, with samples with values of 5 or greater used for library preparation. Using the Kapa RNA HyperPrep Kit with RiboErase (KR1351), rRNA was depleted starting from 25 to 1000 ng of total RNA by hybridization of rRNA to complementary DNA oligonucleotides, followed by treatment with RNase H and DNase to remove rRNA duplexed to DNA. Samples were then fragmented using heat and magnesium. First-strand synthesis was performed using random priming. Second-strand synthesis incorporated dUTPs into the second-strand cDNA. Adapters were then ligated and the library amplified. Strands marked with dUTPs were not amplified, allowing for strand-specific sequencing. Indexed libraries were quantified by qRT-PCR using a commercially available kit (KAPA Biosystems, Roche, Basel, Switzerland) and insert size distribution determined with the LabChip GX or Agilent Bioanalyzer. Samples with a yield of ≥0.5 ng/uL were used for sequencing. Sample concentrations were normalized to 1.2 nM and loaded onto an Illumina NovaSeq flow cell at a concentration that yielded at least 30 million passing filter clusters per sample. Samples were sequenced using 150 bp paired-end sequencing on an Illumina NovaSeq according to Illumina protocols. Data generated during sequencing runs were simultaneously transferred to a high-performance computing cluster. A positive control (prepared bacteriophage Phi X library) provided by Illumina was spiked into every lane at a concentration of 0.3% to monitor sequencing quality in real time. Signal intensities were converted to individual base calls during a run using the system’s Real Time Analysis (RTA) software. Fastq files were quality checked for data yield and base quality. RNA-Seq reads were quality filtered and trimmed using fastp, then aligned to the *Mesocricetus auratus* reference genome using HISAT2 [36]. Quality control of the data was performed by Picard, and StringTie [37] and Ballgown [38] were used to generate normalized expression counts. DESeq2 [39] was used to determine differentially expressed genes (DEGs). To account for variables introduced by viral infection and days post-infection, differential expression analyses was performed by matching each experimental condition with corresponding mock-treated samples (n = 4). Data were further filtered using various combinations of adjusted *p* value (padj) and log2Fold change (log2FC), and DEG subsets organized into cellular pathways using REACTOME pathway analysis [40]. Genes were mapped to human homologues.

### 2.7. Lung Cytokine Expression

The entire right middle lung lobe of each animal was submerged in 1 mL Trizol (Waltham, MA, USA) and total RNA prepared as described above. One μg of total RNA was reverse-transcribed by using SuperScript™ III Reverse Transcriptase (Thermo Fisher, Waltham, MA, USA). Target genes (TNF-α, IL-6, IFN-α, IFN-γ, Cxcl10, Mx2, Ccl2, Ccl8, Ace2) were selected on the basis of their alteration in post-mortem lung from COVID-19 patients [41]. Viral (SARS-CoV-2 RdRp, pH1N1 M gene) and cytokine primer sequences were derived from published papers [14,42,43] or were created according to genomic information (ACE2, NCBI Gene ID: 101823817) and IFN-α; [42]. Amplification efficiency for two sets of primers for each target were tested by using serial dilutions, before selecting one set of primers for each target (Appendix A). Real-time PCR analysis was performed with a CFX Connect Real-Time PCR system by using SYBR Green technology (BioRad, Hercules, CA, USA) and the following thermal cycling conditions: 1 cycle at 95 °C for 30 s followed by 39 cycles of 95 °C for 5 s and 60 °C for 30 s followed by 95 °C for 5 s. After normalization to β-actin expression, data were expressed as relative fold change following analysis with the 2^−ΔΔCt^ method [44].

### 2.8. Statistics

Given the two-factor factorial design (influenza absent or present and SARS-CoV-2-CoV absent or present), a total of 24 hamsters provided 6 hamsters per cell. By using PASS sample size software v15 [45], for a within-cell standard deviation of 0.75, this design achieves 87% power when an F test is used to test both influenza and SARS-CoV-2 at a 5% significance level (an effect size of 0.667) and achieves 100% power when an F test is used to test the influenza and SARS-CoV-2 interaction at a 5% significance level (an effect size of 1.333). To analyze outcomes by days post-infection (dpi), linear mixed effects models were computed that included infection group (i.e., each of influenza and SARS-CoV-2 present versus absent), dpi, sex and an infection group by dpi interaction. For repeated observations (body weight and respiration), a first-order auto-regressive covariance matrix was included, to account for within hamster correlations. To analyze outcomes by sex, models were computed that included infection group, sex and an infection group by sex interaction with correlations from repeated measurement across time accounting for body weight and respiration, as discussed above. Descriptive statistics, including unadjusted comparisons and data visualization, were performed using GraphPad Prizm 10.1.2, and multivariable models were performed using SAS/STAT^®^ 9.4.

## 3. Results

### 3.1. Clinical Phenotype (Experiment 1)

Apart from temporary moderate tachypnea and ruffled fur, animals in all infection groups appeared clinically normal throughout. Control animals and H1N1-inoculated animals of both sexes gained weight over the disease course (Figure 2 and Appendix A). Consistent with prior reports [46,47,48,49], SARS-CoV-2-infected animals experienced significant weight loss with a nadir at 6–8 dpi. As previously noted [50], males experienced significantly greater weight loss. In 3 h coinfected hamsters, weight loss in both sexes approximated the pattern seen in SARS-CoV-2-only infection but diverged at 4 dpi and recovered significantly by 6 dpi, suggesting a delayed protective effect imparted by H1N1 infection. H1N1 given 48 h prior to SARS-CoV-2 almost completely abrogated SARS-CoV-2-typical weight loss in both sexes, beginning at 2 dpi, and increasing to highly significant protection (*p* < 0.001) from 5 dpi onwards. Respiratory rates remained constant throughout disease course in control and H1N1-infected animals of both sexes. In SARS-CoV-2 only and both H1N1/SARS-CoV-2 coinfection groups, tachypnea was evident at 4–7 dpi, with declining respiratory rates by 8 dpi in all groups. Males experienced significantly greater weight loss than females in all infection types, particularly in the 3 h coinfection group, where the delayed protective effect of influenza infection was enhanced in females. Males also experienced significantly greater tachypnea than females in the SARS-CoV-2-only and 3 h coinfection groups (Appendix A).

### 3.2. Pathology and Immunohistochemistry (Experiment 1)

Grossly evident subpleural inflammatory nodules appeared in lungs by 7 dpi in all SARS-CoV-2 cohorts, most evident in SARS-CoV-2-only infected animals. Inflammatory nodules were still evident at 10 dpi in all SARS-CoV-2-infected groups; however, these were greatly reduced in the 48 h dual-infected group. H1N1-infected lungs appeared grossly normal through disease course. Extra-respiratory gross and histopathology was minimal across all infection cohorts and did not differ by sex.

#### 3.2.1. H1N1pdm09 Single Infection

H1N1 infection in our animals caused mild airway-centered interstitial pneumonia [13], evident from 2 to 7 dpi (Figure 3 and Appendix A), resolving by 10 dpi. These changes were qualitatively similar to those reported in fatal influenza infections in humans [51,52] but were much milder. As previously noted [14], H1N1 immunostaining was most robust in the airway epithelium, with lower expression in alveolar epithelial cells. It was weakly discernable or absent after 4 dpi (Figure 4).

#### 3.2.2. SARS-CoV-2-only infection

Consistent with prior reports [13,14,49,53], SARS-CoV-2 infection induced intense nodular broncho-interstitial pneumonia, peaking at 4–7 dpi (Figure 3 and Appendix A). Consistent with prior reports [53], despite striking (but transient) vascular pathology, the endothelial expression of SARS-CoV-2 was exceedingly rare, and no thromboses were noted. Airway pathology in SARS-CoV-2-infected animals peaked slightly later at 2–4 dpi compared to H1N1 infection, resulting in significantly higher histopathology scores in H1N1-infected hamsters at 2 dpi (Figure 3 and Figure 4, Appendix A). After 4 dpi, worsening parenchymal pathology in the SARS-CoV-2-only group resulted in much more severe histopathology scores (*p* < 0.001) at 7–10 dpi (Figure 3 and Figure 4, Appendix A). SARS-CoV-2 NP was detected at 2 dpi in the airway epithelium, expanded to include large areas of the alveolar epithelium at 4 dpi and was greatly diminished at the nadir of pulmonary inflammation at 7 dpi (Figure 4), consistent with viral clearance at 5 dpi, as noted in prior reports [53].

#### 3.2.3. H1N1 and SARS-CoV-2 Coinfections

Components of H1N1 and SARS-CoV-2 typical infection were evident in both coinfection groups (Figure 3). In 48 h coinfected hamsters, SARS-CoV-2 typical nodular broncho-interstitial pneumonia was markedly reduced (Figure 3), with significantly lower corresponding total histopathology scores at 7–10 dpi (*p* < 0.001). Notably, broncho-interstitial pathology appeared in the 48 h coinfection group significantly earlier (*p* < 0.05 at 2 dpi) than in any other SARS-CoV-2 group, consistent with a tendency for H1N1 to elicit early bronchial pathology (Figure 4 and Appendix A). In 48 h coinfected animals, SARS-CoV-2 NP immunostaining was less prevalent throughout the disease course (Figure 4). Interestingly, this immunohistochemical pattern has been previously noted [14], when H1N1 was given 1 day before SARS-CoV-2 in hamsters. In 3 h coinfected hamsters, nodular inflammation was less clearly defined, with H1N1-typical congestion and mild interstitial pneumonia affecting intervening parenchyma (Figure 3), thus resulting in higher (but not significantly so) total histopathology scores (Figure 4). In these animals, SARS-CoV-2 NP immunostaining was similar to that seen in SARS-CoV-2 singly infected animals (Figure 4).

Distinct differences in nasal pathology were noted between H1N1 and SARS-CoV-2 infections (Appendix A). In nasal turbinates, H1N1 immunostaining was most evident in anterior portions of the nose in non-ciliated and ciliated respiratory epithelium. The associated nasal pathology was minimal. H1N1 antigen was evident in cilia and apical cytoplasm consistent with sialic acid distribution [54] and within discrete intracellular structures consistent with lysosomes [55], concentrated near the cytoplasmic membrane (Figure 4). In contrast, SARS-CoV-2 exhibited widespread tropism for regions lined by the olfactory epithelium. As previously noted [56], SARS-CoV-2 NP was detected predominantly in sustentacular-supporting cells of the olfactory epithelium, accompanied by the widespread destruction of the olfactory epithelium (Appendix A). The presence of SARS-CoV-2 NP in the respiratory epithelium overlapped with the pattern of H1N1 immunostaining but was less widespread at these locations. In contrast to the apical distribution seen with H1N1, strong SARS-CoV-2 immunostaining was evident throughout cell bodies of the respiratory epithelium (Figure 4 and Appendix A). In both the olfactory and respiratory epithelium, vigorous migration of Aif+ macrophages into epithelia was evident (Appendix A).

### 3.3. Immune Cell Populations in Bronchoalveolar Lavage Fluid (BALF) and Blood (Experiment 1)

Consistent with other observations [57], the immune cell composition varied by compartment following respiratory infection. BALF granulocytes (known to be primarily composed of neutrophils in Syrian hamsters) proportionally declined in all infection groups from 2 to 7 dpi compared to control animals (Figure 5 and Appendix A), after which they recovered in all groups except the 3 h coinfection group. Because the latter change was accompanied by relative granulocytosis in blood, and histopathologic observation of neutrophils in pulmonary parenchyma in the most severely affected SARS-CoV-2 groups (SARS-CoV-2 alone and 3 h coinfected groups; Figure 3), we interpreted this change to reflect neutrophil consumption in the lung. Of all measures, the reduction in and recovery of neutrophil proportions were most closely associated with histopathologic severity across the disease course, with the 3 h dual infection group experiencing the most marked deviation from normal, and the 48 h group experiencing more rapid normalization of neutrophil dynamics.

Alterations in monocyte/macrophage subsets were evident in all infection groups but were most marked in SARS-CoV-2 groups, consistent with their greater relative pulmonary inflammation compared to H1N1-infected hamsters (Figure 5). There was an enhanced recruitment of monocyte/macrophage populations expressing Aif+ to BALF that persisted throughout the disease course in 3 h dual-infected hamsters. Populations with antigen-presenting capacity (IEk+) were also more heavily recruited to BALF in coinfected groups. When the three populations variously expressing both markers are considered, the protected 48 h dual infection group experienced significantly earlier (by 2 dpi) and greater recruitment of Aif- IEk+ populations to BALF, with a significantly greater presence of this population at later time points (7 and 10 dpi) in peripheral blood. Therefore, prior exposure to H1N1 appeared to enhance the recruitment of populations with antigen-presenting capacity.

### 3.4. Nasal Transcriptomics (Experiment 2)

To explore the protective effect evoked by H1N1, we focused our next experiment on the 48 h dual infection model that had experienced the most protection. We performed RNA sequencing (Appendix A) on the nasal epithelium of H1N1, SARS-CoV-2 and 48 h dual-infected animals, focusing on the early post-infection period (days 1 and 2) before lung pathology or weight loss became evident. In this experiment, only males were used, on the basis of their more severe clinical phenotype and to isolate virus effects from sex influence. Both single SARS-CoV-2 and dual H1N1/SARS-CoV-2 infections induce a greater number of DEGS than H1N1 infection alone (Figure 6; all DEGs padj < 0.05). Within the H1N1 group, nasal gene expression increased significantly between days 1 and 2 post-infection; the overwhelming majority of these genes were upregulated and mapped to pathways involved in innate immunity, responses to viral infection, apoptosis and activation of the inflammasome (Appendix A). Within the infection group, singly infected SARS-CoV-2 animals experienced a significant downregulation of nasal gene expression from day 1 to day 2, while dual-infected animals experience sustained differential expression of a greater number of genes across days 1 and 2 than singly infected SARS-CoV-2 animals. Next, the DEG subset noted above (padj < 0.05) was filtered by an additional condition (differentially expressed in at least three of the six infection groups with a padj < 0.001). This resulted in 256 largely overexpressed genes enriched for innate immune pathways (Appendix A and Figure 6). These were expressed at significantly higher levels in single SARS-CoV-2 and dual H1N1/SARS-CoV-2 groups compared to H1N1 singly infected animals at both time points. Expression was not significantly different between SARS-CoV-2 and dual H1N1/SARS-CoV-2 groups. As noted above, within the H1N1 group, nasal gene expression increased significantly between days 1 and 2. In contrast to single SARS-CoV-2 infection, significant downregulation of DEGs was evident by day 2 in dual-infected animals. Because dual-infected animals receive SARS-CoV-2 48 h after H1N1 infection, upregulated gene expression induced by H1N1 infection is already evident at the time of SARS-CoV-2 inoculation. These include many genes engaged in innate immune and antiviral pathways (Figure 6 and Supplementary Data S2 and S3) that are already engaged by the time the host is exposed to SARS-CoV-2. When differential gene expression was compared between dual-infected animals and singly SARS-CoV-2 infected hamsters, the vast majority of DEGs occurred at day 1 (1130 DEGS), and most of these (917 genes) were relatively underexpressed in dual-infected animals (Appendix A). Underexpressed genes mapped to pathways associated with transcription, translation and mitochondrial function, as well as several pathways involved in cellular responses to viral infection (Figure 6). When comparing DEG in both SARS-CoV-2-infected groups at day 2 (i.e., 2 days after SARS-CoV-2-only infection and 4 days after H1N1 infection in the dual-infected group), only five known genes (Cd3d, Ctsw, Kirrel2, Osbpl1a and Aurkb) were differentially expressed at padj < 0.05 (Appendix A).

### 3.5. Viral and Cytokine Gene Expression (qRT-PCR)

Consistent with nasal transcriptome data, nasal H1N1 M gene expression significantly increased between days 1 and 2 in H1N1 singly infected hamsters (Figure 7). In contrast, declining nasal H1N1 M gene expression was evident by day 2 in 48 h dual-infected animals, with significantly lower H1N1 expression at this time point compared to singly H1N1-infected animals. In the lung, the temporal expression of H1N1 M gene expression in H1N1 singly infected hamsters closely resembled that of the histopathologic course of the disease (Figure 4), peaking at 4 dpi. In the 48 h dual-infected animals, peak H1N1 M gene expression occurred at 2 dpi. As the dual-infected group would have been 4 days post H1N1 infection at day 2 post SARS-CoV-2 infection, these findings are unsurprising. Nasal SARS-CoV-2 RdRp expression was comparable at both time points in SARS-CoV-2 singly infected and dual-infected groups. In the lung, SARS-CoV-2 RdRp expression in SARS-CoV-2 singly infected animals peaked at 2–4 dpi, prior to the clinical and histopathologic nadir of disease at 7 dpi. In the 48 h dual-infected group, SARS-CoV-2 RdRp expression was lower at all time points, reaching statistical significance at 2 dpi. In singly infected animals, significantly higher histopathology scores in SARS-CoV-2 compared to H1N1-infected animals (Appendix A) were accompanied by SARS-CoV-2 viral loads that were approximately one order of magnitude higher than H1N1 viral load. Higher respective viral loads in SARS-CoV-2 compared to H1N1 singly infected hamsters have been previously noted [13], thus implicating viral load as a contributory factor in disease severity. No significant sex differences were found for histopathology scores or viral gene expression.

Similar to the pattern noted in nasal transcriptome data, lung cytokine expression was significantly higher in singly infected SARS-CoV-2 animals compared to singly infected H1N1 animals (Figure 7). Significant differences were largely confined to the 4 dpi time point, except for M × 2 where they were evident earlier (2 dpi) and Ccl2, where they persisted for longer (7 dpi). Compared to the singly infected SARS-CoV-2 group, the dual-infected group experienced lower cytokine expression levels. These were also clustered around the 4 dpi time point, except for M × 2 where they were evident earlier (2 dpi). IFN-γ expression was relatively higher at 2 dpi in the dual-infected group (comparable to expression 4 dpi in singly infected H1N1 hamsters) compared to singly infected animals; however, this did not reach statistical significance. In all SARS-CoV-2-infected cohorts, the temporal expression pattern of a cluster of chemokines/cytokines (IL-6, Cxcl-10, Ccl-8, Ccl-2 and IFN-γ) mimicked the temporal pattern of viral load, with the expression of both peaking at 4 dpi (Figure 6 and Figure 7). This relationship between viral replication and chemokine/cytokine induction agrees with in vitro [25] and in vivo data consistent with the viral clearance and appearance of IgM by 5 dpi in SARS-CoV-2-infected hamsters [53]. Consistent with prior reports [53], M × 2 was an early responder, peaking at 2 dpi, declining before viral loads peaked. IFN-α levels oscillated over time in both coinfected groups but generally stayed low. The induction of Ace-2 was not seen in any cohort, although its expression was significantly higher at 2 dpi in SARS-CoV-2-only animals compared to all other infection groups (Appendix A).

Singly SARS-CoV-2-only infected males experienced a tendency towards a higher expression of IL-6, Cxcl-10, Ccl-8, Ccl-2 and IFN-γ. This was significant for CxCl-10 (*p* = 0.02) and IL-6 (*p* = 0.009), implicating these in greater clinical disease severity (Appendix A). Female SARS-CoV-2-only infected animals expressed significantly higher levels of IFN-α and Ace-2. TNF-α was significantly higher in female 48 h coinfected hamsters. Overall, the expression of IFN-α, Ace-2 and TNF-α was variable and rather low overall. While TNF-α is often elevated in inflammatory conditions, studies describe TNF-α as downregulated in SARS-CoV-2-infected hamsters [53] or variably upregulated in both SARS-CoV-2 and influenza infection [58].

## 4. Discussion

In this study, we characterized clearly distinguishable mild (caused by H1N1) and more severe (caused by SARS-CoV-2) disease in animals given single infections. We established that H1N1 given prior to SARS-CoV-2 largely mitigated SARS-CoV-2-associated disease when given 48 h prior, and elicited hybrid responses when given 3 h prior. We focused further transcriptomic analyses on nasal tissues using new cohorts of singly infected and 48 h dual-infected male animals and extended observations to the lung tissue of the original cohorts of both sexes.

Consistent with human clinical data [59], H1N1 caused a milder clinical phenotype than SARS-CoV-2. The clinical and pulmonary histopathologic features of H1N1 in our study resembled those previously described in hamsters [14,54] and ferrets [60]. We reproduced nodular pneumonia in SARS-CoV-2-infected hamsters, as reported by others [46,47,61]. Consistent with the prevailing view that females are able to counter viral infections more effectively [62,63], in both singly infected groups, males experienced significantly greater weight loss and tachypnea, particularly following SARS-CoV-2 infection. By varying the timing of H1N1 infection, we were able to profoundly modulate the SARS-CoV-2 phenotype. In the 3 h coinfection group, we detected a hybrid phenotype. Compared to worse clinicopathologic outcomes following simultaneous influenza/SARS-CoV-2 coinfections in hamsters [13,14], ferrets [12] and mice [64], animals in our 3 h dual-infected group experienced weight loss similar to the SARS-CoV-2-only group until 4 dpi, at which point clinical protection characterized by accelerating weight gain was seen. However, 3 h dual-infected animals developed slightly worse pulmonary inflammation (*p* = 0.05 at 7 dpi), accompanied by more profound neutrophil alterations in BALF and blood, and higher relative increases in macrophage proportions in BALF. H1N1 given 48 h prior to SARS-CoV-2 resulted in remarkable mitigation of the SARS-CoV-2 clinical and pathologic phenotype. Compared to SARS-CoV-2 singly infected hamsters, the 48 h dual-infected animals experienced more rapid normalization of granulocyte dynamics in BALF and blood. This was accompanied by an enhanced early recruitment of antigen-presenting myeloid populations to BALF, with a significantly greater presence of this population at later time points in peripheral blood. This is consistent with enhanced the MHCII expression characteristic of influenza infection [65] and suggests that an increased induction of early APC capacity by prior H1N1 infection may contribute to protection.

Immune priming in nasal mucosa by prior H1N1 infection is likely to be an important mechanism underlying protection against subsequent SARS-CoV-2 infection in the 48 h cohort. In animal studies, protection against subsequent viral infection has been achieved with intranasal administration of live attenuated influenza vaccine in SARS-CoV-2-infected ferrets [28], pattern recognition receptor (PRR) ligands in SARS-CoV-2-infected mice [66], interferon-α in SARS-CoV-2-infected hamsters [67] and polyinosinic/polycytidylic acid in SARS-CoV-2 and influenza-infected mice [68]. We and others [56,69] demonstrate widespread nasal SARS-CoV-2 expression at 2–4 dpi. Additionally, we demonstrate a robust infiltration of Aif+ macrophages into infected nasal epithelia and the colocalization of H1N1 and SARS-CoV-2 in the nasal respiratory epithelium (as well as broncho-alveolar epithelium and intra-alveolar macrophages). In hamsters, SARS-CoV-2 given with or prior to H3N2 is able to repress influenza expression in the nose and lung [70]. In airway epithelial organoids [25] and airway epithelial cells [26,27], prior rhinoviral infection blocked the replication of SARS-CoV-2 given 1 to 3 days later. This was accompanied by an accelerated induction of IFNλ1, CxCl10 [25] and other ISGs in both infected and uninfected bystander cells.

To further explore the protective effect evoked by H1N1, we focused our next experiment on nasal transcriptomics using the 48 h dual infection model that had experienced the most protection. Compared to both single SARS-CoV-2 and dual H1N1/SARS-CoV-2 groups, H1N1 infection evoked lower DEG expression overall, with a sharp increase in predominantly upregulated genes engaged in innate immune and antiviral pathways by the second day of infection. Therefore, by the time animals in the 48 h dual infection group received their SARS-CoV-2 inoculation, their nasal environment was already primed to express a broad array of interferon-stimulated genes (ISGs), as noted by other groups [71]. Notable upregulated genes included Ddx58 (otherwise known as retinoic acid-inducible gene I, or RIG-I) and Dhx58 (or RIG-I-like receptor LGP2). These are key mediators of innate antiviral responses that are rapidly upregulated by influenza [72,73]. Higher baseline expression of these pattern recognition receptors in the respiratory tract of children results in stronger innate responses and lower disease susceptibility following SARS-CoV-2 infection compared to adults [74]. Early ISG upregulation characterizes viral containment and protective immune trajectories in patients [75]. These immune observations are very much in line with our observations in our protected 48 h dual-infected cohort. At day 1 post-infection with SARS-CoV-2, dual-infected animals experienced profound downregulation of DEGs compared to their singly SARS-CoV-2 infected counterparts. In addition to cellular responses to viral infection, underexpressed genes mapped to pathways associated with transcription, translation, mitochondrial function, apoptosis and cellular starvation. Understanding how these alterations contribute to protection or not in our model is likely to be complex and will require additional studies. Mitochondria are engaged in the antiviral response through the activation of mitochondrial antiviral-signaling protein (MAVS) by cytosolic viral sensors such as RIG-1 [76,77] to induce apoptosis in infected cells [76]. These interactions are influenced by mitochondrial redox status and are generally thought to promote viral replication [78] and contribute to post-COVID-19 mitochondrial dysfunction [79]. In other studies, SARS-CoV-2 is able to inhibit mitochondrial gene transcription and trigger the integrated stress response [80] and associated interferon responses [81]. Differential gene expression between single and dual-infected SARS-CoV-2 groups was very short-lived, and by day 2, only five known genes (Cd3d, Ctsw, Kirrel2, Osbpl1a and Aurkb) were differentially expressed. One of these overexpressed, Cd3d, would support the earlier activation of adaptive responses implied by our flow cytometric data in the 48 h dual-infected group. Interestingly, desmoglein (Dsg3) was the most highly overexpressed gene in the dual-infected group. This may provide mechanistic insight regarding the emergence of pemphigus vulgaris after SARS-CoV-2 and influenza infection or vaccination [82,83,84,85].

Dual-infected animals (male) experienced significantly reduced H1N1 M gene levels in nasal tissue at day 2, compared to H1N1 singly infected groups. When extended to lung samples from both sexes, H1N1 M gene levels were higher at 2 dpi in dual-infected animals (i.e., 4 days after H1N1 infection in this group), and equivalent to H1N1 M gene expression at 4 dpi in H1N1 singly infected animals. In the lung, SARS-CoV-2 RdRp expression was lower at all time points, reaching statistical significance at 2 dpi. Compared to the singly infected SARS-CoV-2 group, the dual-infected group experienced lower cytokine expression levels in the lung, clustered around the 4 dpi time point, except for M × 2 where they were evident earlier (2 dpi). IFN-γ expression was relatively higher at 2 dpi in the dual-infected group (comparable to expression 4 dpi in singly infected H1N1 hamsters), consistent with immune priming by earlier H1N1 infection. Our data confirm that H1N1 given 2 days prior to SARS-CoV-2 induces rapid and broad gene expression in the upper respiratory tract. This is associated with the marked mitigation of SARS-CoV-2 pulmonary disease accompanied by reduced lung SARS-CoV-2 viral and cytokine expression. In 3 h dual-infected animals, accelerated clinical recovery implies that H1N1 may induce some protective gene expression within 3 h but that temporally proximate dual infections do not protect against pulmonary pathology. Conversely, extension of the time between H1N1 infection and SARS-CoV-2 infection may not afford additional protection as antiviral responses to H1N1 rapidly reduce its viral load (by 4 days post-infection), thus reducing the associated innate response that could serve as protection against subsequent DARS-CoV-2 infection. These data imply that clinical outcomes are exquisitely sensitive to the timing of coinfection and may be quite variable in animals [13,64,71,86] and humans [4,5,16,87].

## 5. Conclusions and Limitations

Contrasting animal model and patient reports on the impact of influenza and SARS-CoV-2 coinfection can be reconciled when considering the interaction of these two viruses at the individual and population levels. At a population level, it is likely that endemic SARS-CoV-2 will join the seasonal respiratory virus landscape in which competitive or synergistic viral interactions influence the overall prevalence of respiratory viral diseases each year. Epidemiologic studies have demonstrated that earlier seasonal influenza occurrence resulted in the delayed seasonal occurrence of other respiratory viral infections, particularly respiratory syncytial virus (RSV) [88,89]. We confirm that prior infection with H1N1 mitigates SARS-CoV-2-associated disease within a time frame consistent with innate immune priming. However, in hospitalized patients, coinfection is generally associated with worse outcomes [4,5]. In patients, modifying factors, such as age, sex, socio-demographic characteristics and comorbidity [90,91,92], influence disease severity in both SARS-CoV-2 and influenza infections. Our data should in no way discourage vaccination for either virus. Additionally, hamsters lack the fibrin-rich exudative, proliferative and fibrotic progression characteristic of diffuse alveolar damage (DAD) [32,93,94] in severe SARS-CoV-2 or fatal H1N1 infections in humans [52], and they do not experience multiorgan injury and concurrent thromboembolism and coagulopathy [95] seen in severe COVID-19. Nevertheless, our nasal transcriptomic data reflect the complex interplay between hundreds of ISGs in diverse cell types in the upper respiratory tract that are altered by viral infections [53]. This is a fertile area for future study, and our 48 h coinfection data provide clear time points associated with protection.

## Figures and Tables

**Figure 1 viruses-16-00246-f001:**
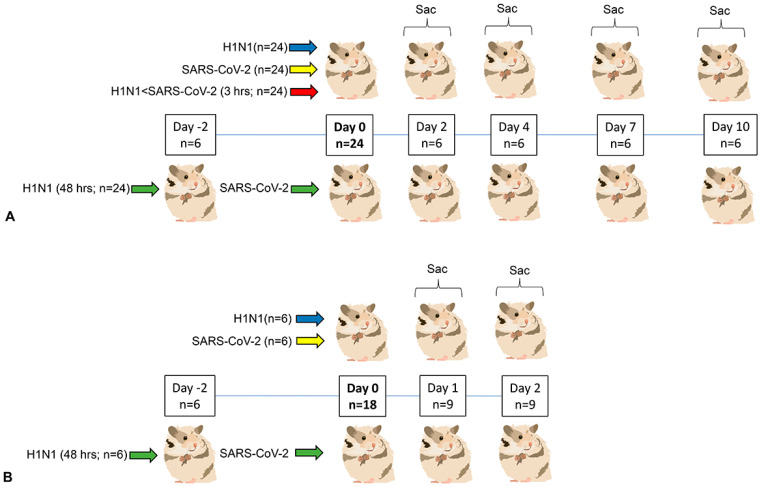
Study design. (**A**) Experiment 1. Four infection groups (24 animal per infection type, split evenly by sex) were studied. Two groups were inoculated intranasally on Day 0 with 10^6^ PFU/animal of either H1N1 (blue) or SARS2 (yellow). In coinfected groups, animals were given H1N1 first, followed by SARS-CoV-2 given either 3 h (red) or 48 h (green) later. Identical doses and volumes were used for each virus in coinfections. Six animals were sacrificed at 2, 4, 7 and 10 days post-Day 0. A 24-animal control cohort was aggregated from 4 to 6 animals accompanying each infection group. Control animals were mock inoculated with media and sacrificed on similar days. (**B**) Experiment 2: Three infection groups (6 animals per infection type, male) were studied. Two groups were inoculated intranasally on Day 0 with 10^6^ PFU/animal of either H1N1 (blue) or SARS2 (yellow). Only one coinfection group (H1N1 followed by SARS-CoV-2 given 48 h later) was used. Half of the animals in each infection group were sacrificed 24 h post Day 0, and the remainder sacrificed 24 h later. Four media-only inoculated animals (male) sacrificed on similar days provided controls.

**Figure 2 viruses-16-00246-f002:**
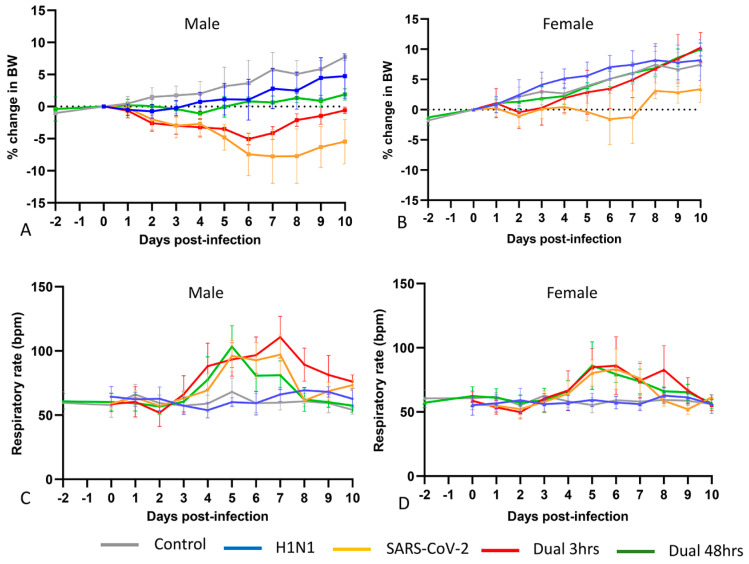
Body weight change and respiratory rate by infection group. (**A,B**): Body weight change in males (**A**) and females (**B**). Mock-inoculated control animals (grey) of both sexes gain weight over 10 days. This pattern was replicated in both sexes given H1N1 (blue), and in the H1N1-SARS-CoV-2 48 h coinfection (green), although the rate of weight gain was slower in males than in females. In SARS-CoV-2-only infected animals (orange), both sexes lost weight, however females regained weight faster. With H1N1-SARS-CoV-2 3 h coinfection (red), weight loss in both sexes approximated that seen in SARS-CoV-2-only infection but recovered by 4–5 dpi. Weight gain in females accelerated to approximate that seen in control females whereas males recover at a slower rate. (**C,D**): Respiratory rate in males (**C**) and females (**D**). Respiratory rates remain constant in control (grey) and H1N1 infected (blue) males and females. In SARS-CoV-2-only and H1N1/SARS-CoV-2 coinfection groups, respiratory rates become elevated by 4–5 dpi, with resolution by 8 dpi in all groups. Males experience higher respiratory rates during this period than females. BW = body weight; bpm = breaths per minute. *p*-values for comparisons across infection type and sex are given in Appendix A.

**Figure 3 viruses-16-00246-f003:**
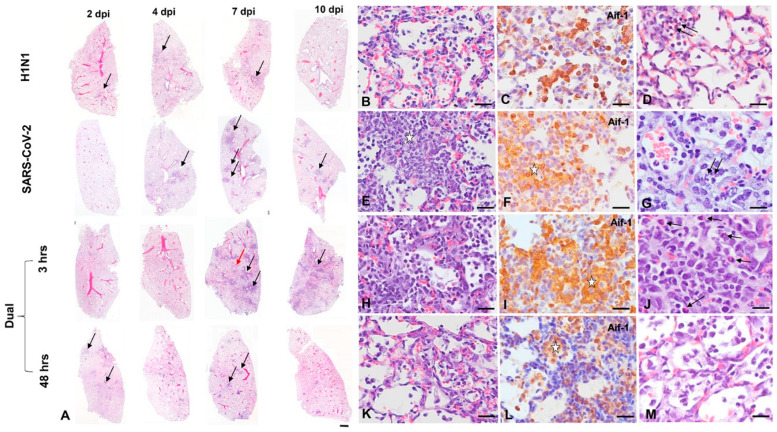
Comparative histopathology of H1N1 only, SARS-CoV-22 only and H1N1/SARS-CoV-2 coinfection groups (**A**) Representative subgross histology, left lung, 2–10 dpi, males only. Mild, rather diffuse interstitial pneumonia (arrows) is evident in H1N1 only infected hamsters from 2–7 dpi. In the SARS-CoV-2 only group, well-defined nodular broncho-interstitial pneumonia reaches its nadir at 7 dpi (black arrows) with partial resolution at 10 dpi. Intervening parenchyma is normal. In the 3 h coinfected group, nodular inflammation (black arrows) with intervening alveolar congestion and mild interstitial inflammation resulting from H1N1 infection (red arrow) is evident. In 48 h coinfected animals, inflammation appears by 2 dpi in some animals (black arrow), but nodular broncho-interstitial pneumonia is less severe throughout disease course. Bar = 500 µm. (**B–D**): H1N1 only infection, 4 dpi, male. Interstitial congestion with mild alveolar consolidation (**B**) is present. Parenchymal consolidation is minimal, and Inflammation dominated by intra-alveolar macrophages (**C**), with few neutrophils (arrow, **D**). (**E–G**): SARS-CoV-2-only infection, 7 dpi, male. Dense interstitial infiltration (asterisk, **E**) characterizes well-defined nodular inflammation. Infiltrating cells are macrophage dominant (asterisk, **F**) with fewer neutrophils (arrows, **G**). (**H–J**): H1N1 < SARS-CoV-2 3 h coinfection, 7 dpi, male. Interstitial and alveolar infiltration (**H**) by macrophage dominant inflammation (**I**) and scattered neutrophils (arrows, **J**) is present. (**K–M**): H1N1 < SARS-CoV-2 48 h coinfection, 7 dpi, male. Nodular lesions are less severe than in SARS2 only infection, with reduced interstitial consolidation (**K**) and predominant intra-alveolar macrophage (**L**) infiltration. Neutrophils are difficult to detect (**M**). Haematoxylin and eosin (**A**,**B**,**E**,**H**,**K**) Aif-1 immunohistochemistry (**C**,**F**,**I**,**L**) Bar = 500 µm (**A**), 20 µm (**B**,**C**,**E**,**F**,**H**,**I**,**K**,**L**); 10 µm (**D**,**G**,**J**,**M**).

**Figure 4 viruses-16-00246-f004:**
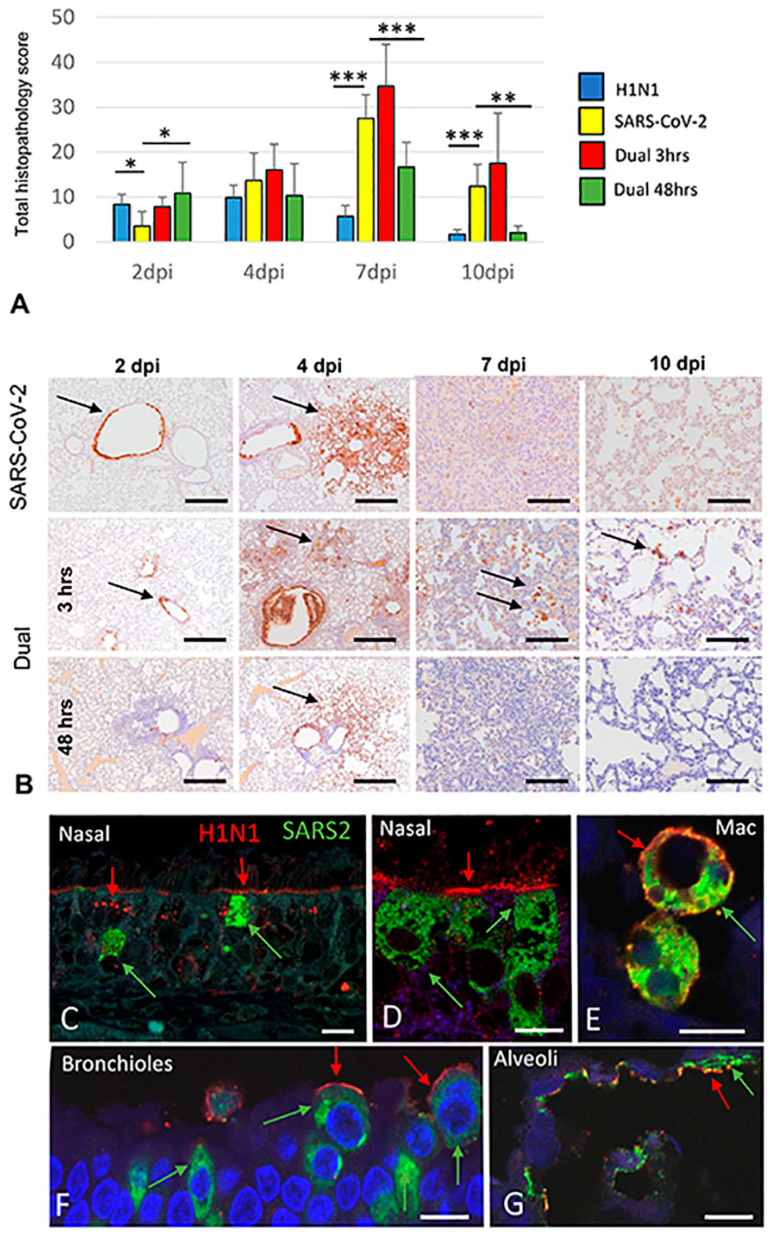
Total histopathology score and viral immunohistochemistry in single and dual-infected hamsters. (**A**) Total histopathology score, H1N1 only, SARS-CoV-2 only and dual coinfection groups. Early onset airway pathology results in higher scores for H1N1 infected animals (blue) at 2 dpi. SARS-CoV-2 associated pathology (yellow) develops more slowly but is more severe with a nadir at 7 dpi. This pattern is exacerbated in the 3 h coinfection group (red) and mitigated in the 48 h coinfection group (green). Sexes are combined. * *p* < 0.05, ** *p* < 0.01, *** *p* < 0.001. Methods for assessing total histopathology score are given in Appendix A. (**B**) Pulmonary SARS-CoV-2 NP immunohistochemistry, single and dual SARS2 infected groups. SARS-CoV-2 NP is evident in bronchiolar epithelium at 2 dpi (arrows), with the highest levels detected in parenchyma at 4 dpi (arrows), prior to the nadir of pulmonary inflammation at 7 dpi in all groups. SARS-CoV-2 NP immunostaining is reduced in the 48 h coinfection group and persists in single cells in the 3 h coinfection group (arrows). (**C**,**D**) Respiratory epithelium, maxillary sinus, dual 3 h coinfection, 4 dpi, male. H1N1 is detected in cilia, apical aspects and within discrete intracellular structures consistent with lysosomes (red arrows). SARS-CoV-2 NP is diffusely distributed through the cytoplasm (green arrows). (**E**) Intra-alveolar macrophages, dual 3 h coinfection, 4 dpi, male. Detection of H1N1 concentrated near cytoplasmic membrane (red arrow) and SARS-CoV-2 NP seen throughout cytoplasm (green arrow) in coinfected intra-alveolar macrophages. (**F**) Bronchiolar epithelium, H1N1 < SARS-CoV-2 3 h coinfection, 4 dpi, male. Apical expression of H1N1 is evident in bronchiolar epithelium (red arrows). SARS-CoV-2 NP is diffusely distributed through the cytoplasm (green arrows). (**G**) Alveolar epithelium, dual 3 h coinfection, 4 dpi, male. Although SARS-CoV-2 NP immunostaining (green) is more widespread than H1N1 immunostaining (red) in alveolar epithelium, colocalization can be seen in some areas (red and green arrow). Bar = 10 µm (**B**): SARS-CoV-2 immunohistochemistry, light microscopy. Bar =100 µm (all panels) (**C–G**): SARS-CoV-2 and H1N1 immunohistochemistry, confocal microscopy. Bar = 20 µm (**E**); 5 µm (**F**,**G**).

**Figure 5 viruses-16-00246-f005:**
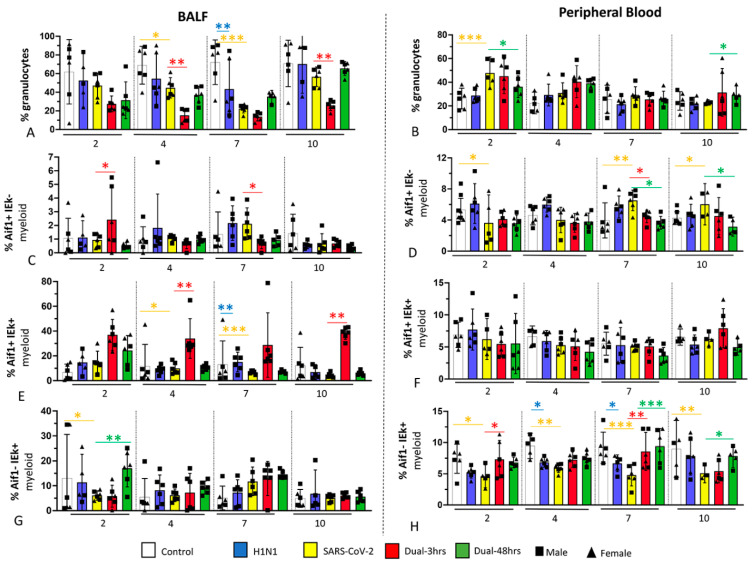
Flow cytometry of myeloid cell populations in bronchoalveolar lavage fluid and blood of infected hamsters at 2, 4, 7 and 10 dpi and the uninfected controls. The cell subsets were defined and gated as described in Methods and Appendix A and the frequencies expressed as a percentage of all viable leukocytes: A, B: Granulocytes in BALF (**A**) and peripheral blood (**B**). C-H: Subsets of non-granulocytic myeloid cells distinguished by their varied expression of Aif1 and IEk; Aif + IEk- myeloid cells in BALF (**C**) and peripheral blood (**D**); Aif+ IEk+ cells in BALF (**E**) and peripheral blood (**F**); Aif- IEk+ myeloid cells in BALF (**G**) and peripheral blood (**H**). Individual values for each animal and means are presented. Numbers of animals for BAL analyses: n = 23 (Control); n = 23 (H1N1); n = 10 (SARS-CoV-2); n = 23 (Dual-3 h); n = 23 (Dual-48 h); for blood analyses n = 23 (Control); n = 24 (H1N1); n = 19 (SARS-CoV-2); n = 24 (Dual-3 h); n = 23 (Dual-48 h). Sexes are combined for statistical analysis. * *p* < 0.05, ** *p* < 0.01, *** *p* < 0.001. Asterisks and bars are colored as follows. Blue (H1N1/Control); Yellow (SARS-CoV-2/Control); Red (Dual-3 h/SARS-CoV-2); Green (Dual-48 h/SARS-CoV-2). See Appendix A for values.

**Figure 6 viruses-16-00246-f006:**
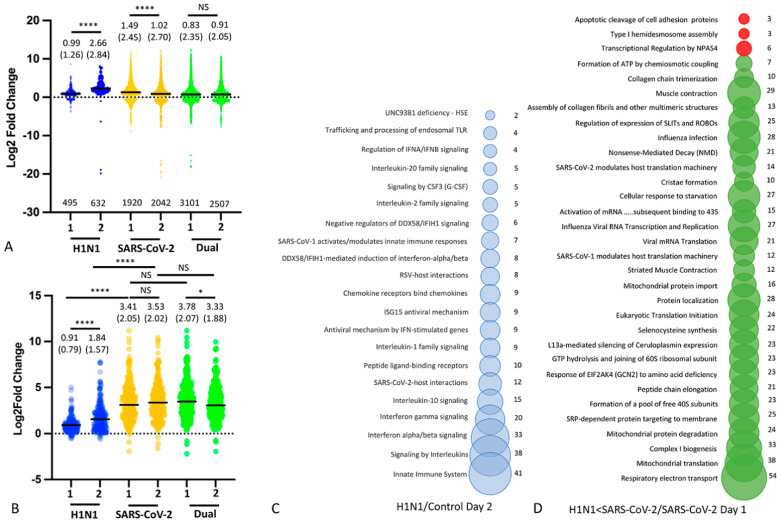
Differentially expressed genes (DEGs) in nasal epithelial gene expression at day 1 and 2 post-inoculation (H1N1, SARS-CoV-2 single infections, and 48 h H1N1/SARS-CoV-2 dual infection). (**A**) Violin plot illustrating expression levels (Log2Fold Change) of DEGS (padj < 0.05) in each infection group (n = 3 males per infection type and day after infection, noted as 1 or 2 on X axis), compared to the same control group (mock inoculated males, n = 4). Mean (standard deviation) for each group given above its violin plot; total number of DEGs for each group (Blue: H1N1, Yellow: SARS-CoV-2; Green: Dual infection) is given below. Significance of mean gene expression between pairs of infection groups was assessed with a *t*-test, with a Bonferroni correction (0.05/3 comparisons =0.0166). **** (*p* < 0.0001), * (*p* < 0.05), NS = not significant. (**B**) Violin plot illustrating expression levels (Log2Fold Change) of DEGS shown in A, filtered by an additional condition (differentially expressed in at least three of the six infection groups with a padj < 0.001). Mean (standard deviation) for each group (Blue: H1N1, Yellow: SARS-CoV-2; Green: Dual infection) given above its violin plot. The full list of genes (n = 246 genes) is given in Appendix A. Significance of mean gene expression between pairs of infection groups was assessed with a *t*-test, with a Bonferroni correction (0.05/3 comparisons =0.0166). **** (*p* < 0.0001), * (*p* < 0.05), NS = not significant. (**C**) Bubble graph illustrating REACTOME pathway analysis of significantly differentially expressed genes (n = 158, padj > 0.05) at day 2 in H1N1 (blue) singly infected hamsters, within the 246 gene subset of DEGS shown in B. Pathways engaged with an Entities pVal < 0.05 are illustrated, with the number of genes within each pathway given to the right of each bubble. Of 158 genes used for analysis, 34 failed to map to human homologues. (**D**) Bubble graph illustrating REACTOME pathway analysis of significantly differentially expressed genes (padj > 0.05) at day 1 in H1N1 < SARS-CoV-2 dual-infected hamsters compared to SARS-CoV-2 singly infected hamsters. Pathways engaged with an Entities pVal < 0.05 are illustrated, with the number of genes within each pathway given to the right of each bubble. Underexpressed pathways are colored green, overexpressed pathways colored red. Of 1130 genes used in REACTOME analysis, 374 did not map to human homologues.

**Figure 7 viruses-16-00246-f007:**
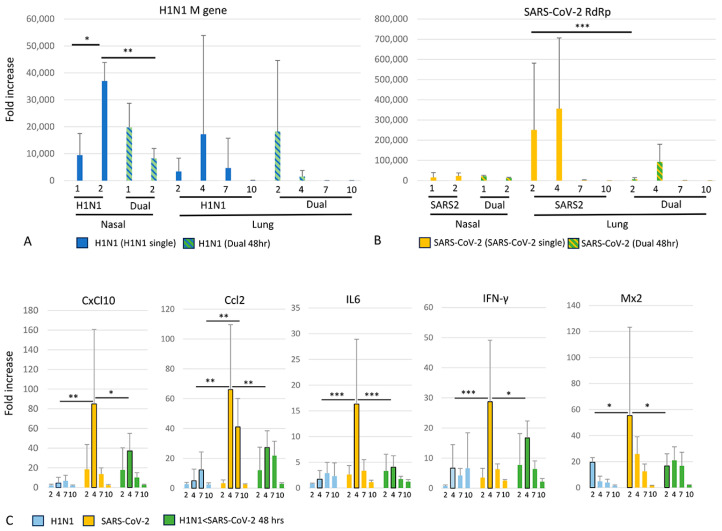
Viral gene expression (nasal and lung) and cytokine expression (lung) in H1N1, SARS-CoV-2 singly infected, and 48 h dual-infected groups at the dpi indicated. (**A**) H1N1 M gene expression in nasal and lung tissue in H1N1 and dual-infected groups. (**B**) SARS-CoV-2 RdRp expression in nasal and lung tissue in SARS-CoV-2 and dual-infected groups. (**C**) Selected lung cytokine expression in H1N1, SARS-CoV-2 singly infected, and 48 h dual-infected groups. Columns associated with significant comparisons are outlined in black. Expression of some cytokines (TNF-α, IFN-α, Ace-2, Ccl-8) are not illustrated, however full viral and cytokine expression data are given in Appendix A. qRT-PCR, * *p* < 0.05, ** *p* < 0.01, *** *p* < 0.001.

## Data Availability

RNA sequencing data were uploaded through GEO (https://www.ncbi.nlm.nih.gov/geo/; Accession number GSE254516), and remaining supporting data are given in the Appendix A.

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
