# Peer review of "Prior Influenza Infection Mitigates SARS-CoV-2 Disease in Syrian Hamsters"

_viruses, 2024, doi:10.3390/v16020246_

Round 1

Reviewer 1 Report

Comments and Suggestions for Authors

This study investigates the interaction between SARS-CoV-2 and H1N1 influenza in Syrian hamsters, finding that H1N1 infection 48 hours before SARS-CoV-2 mitigates weight loss and lung pathology. The protective effects involve rapid upregulation of innate and antiviral pathways, leading to reduced SARS-CoV-2 viral load and lower cytokine levels. These findings suggest a potential influence on the seasonal infection rates of these viruses, emphasizing the complex interplay in the host's immune response. However, the article mentions that H1N1 infection can rapidly upregulate immune and antiviral pathways, yet it does not provide detailed insights into how these pathways function during SARS-CoV-2 infection. In-depth research into the immune regulatory mechanisms could contribute to unraveling the detailed mechanisms of virus interactions.

Comments on the Quality of English Language

The quality of English language is fine. There are some minor grammar errors or expressions that are not quite clear. For example, "Dual infected animals also experienced significant transient downregulation of mitochondrial and viral replication pathways." This sentence may cause some ambiguity in terms of the subject. A clearer expression could be: "The animals that were infected with both viruses also showed a notable and temporary downregulation of mitochondrial and viral replication pathways." For example,"By quantitative RT-PCR, we confirmed reduced SARS-CoV-2 viral load and lower cytokine levels throughout disease course in lung of dual infected animals." - This sentence could be expressed more clearly as: "Quantitative RT-PCR confirmed a decrease in the SARS-CoV-2 viral load and lower cytokine levels in the lungs of animals infected with both viruses throughout the course of the disease."

Author Response

We thank the reviewer for their very timely review and helpful comments. We have addressed these fully.

However, the article mentions that H1N1 infection can rapidly upregulate immune and antiviral pathways, yet it does not provide detailed insights into how these pathways function during SARS-CoV-2 infection. In-depth research into the immune regulatory mechanisms could contribute to unraveling the detailed mechanisms of virus interactions.

Response: We have expanded the discussion as follows: Page 16, line 600-606:

Notable upregulated genes included Ddx58 (otherwise known as retinoic acid-inducible gene I, or RIG-I) and Dhx58 (or RIG-I-like receptor LGP2). These are key mediators of innate antiviral responses that are rapidly upregulated by influenza [72, 73]. Higher baseline expression of these pattern recognition receptors in the respiratory tract of children results in stronger innate responses and lower disease susceptibility following SARS-CoV-2 infection compared to adults [74]. Early ISG upregulation characterizes viral containment and protective immune trajectories in patients [75]. These immune observations are very much in line with our observations in our protected 48-hour dual infected cohort.”

Page 17, line 609-616: “Understanding how these alterations contribute to protection or not in our model is likely to be complex and will require additional studies. Mitochondria are engaged in the antiviral response through activation of mitochondrial antiviral-signaling protein (MAVS) by cytosolic viral sensors such as RIG-1 to induce apoptosis of infected cells [76]. These interactions are influenced by mitochondrial redox status and are generally thought to promote viral replication [78] and contribute to post-COVID mitochondrial dysfunction [79].”

The quality of English language is fine. There are some minor grammar errors or expressions that are not quite clear. For example, "Dual infected animals also experienced significant transient downregulation of mitochondrial and viral replication pathways." This sentence may cause some ambiguity in terms of the subject. A clearer expression could be: "The animals that were infected with both viruses also showed a notable and temporary downregulation of mitochondrial and viral replication pathways." For example,"By quantitative RT-PCR, we confirmed reduced SARS-CoV-2 viral load and lower cytokine levels throughout disease course in lung of dual infected animals." - This sentence could be expressed more clearly as: "Quantitative RT-PCR confirmed a decrease in the SARS-CoV-2 viral load and lower cytokine levels in the lungs of animals infected with both viruses throughout the course of the disease."

Response:  Thank you – these suggestions have been adopted (Abstract)

Reviewer 2 Report

Comments and Suggestions for Authors

In this study, Pietro et al conducted well-controlled experiments to investigate the potential clinical impact of HINI coinfection on a following SARS-COV-S exposure. To this end, they first established the exp 1 including multiple infection time courses to determine the most effective co-infection strategy. The data showed that compared to a nearly same-time co-infection, an earlier pre-exposure to H1N1 provided a better clinical benefit. It would be interesting to analyze a wider range of post-influenza time points before SARS-CoV-2 inoculation.

The following works focus on elucidating the potential mechanism underlying this phenomenon and found that pre-infection with H1N1 induced rapid upregulation of innate immune and antiviral genes in the nasal epithelium. At the time of SARS-CoV-2 infection 48 hours later, the nasal environment was already primed with expression of interferon-stimulated genes reflecting a more protective tissue immunity. The results suggest innate immune training by influenza infection rapidly primes antiviral defenses that mitigate SARS-CoV-2 disease. Also, whether a pro-longed gap between two infections would involve in more immune components that might provide additional benefit to protect against SARS-COV2?

Overall, this is a nice study demonstrating heterologous protection against SARS-CoV-2 by recent influenza infection in hamsters. The manuscript is well-written and presents some interesting findings on viral interference between influenza and SARS-CoV-2. No major revisions appear needed. With minor revisions, the manuscript should make a good contribution to the literature.

Some other suggestions:

1.       In fig5, the symbol format is inconsistent with the legend.

2.       It would be good to discuss the implications of these findings for population-level seasonal dynamics of influenza and SARS-CoV-2 co-circulation.

Author Response

Response to Reviewer 2

We thank the reviewer for their very timely review and helpful comments. We have addressed these fully.

Response: The question whether an increased gap between the two infections is addressed in the discussion as follows:

Page 17, line 645-649: Conversely, extension of the time between H1N1 infection and SARS-CoV-2 infection may not afford additional protection as antiviral responses to H1N1 rapidly reduce its viral load (by 4 days post infection) thus reducing the associated innate response that could serve as protection against subsequent DARS-CoV-2 infection.”

The implications of these findings for population-level seasonal dynamics of influenza and SARS-CoV-2 co-circulation

Response: We have expanded the conclusions and limitations section to address this as follows:

Page 17, 658-660:  Epidemiologic studies have demonstrated that earlier seasonal influenza occurrence resulted in delayed seasonal occurrence of other respiratory viral infection, particularly respiratory syncytial virus (RSV) (88, 89)

Given the propensity for dual infections to result in worse outcomes in patients, we have also emphasized that people should still vaccinate against both viruses.

Page 18, line 665 “our data should in no way discourage vaccination for either virus”.

Response:  In fig5, the legend has been corrected to align with the symbol format.  

Reviewer 3 Report

Comments and Suggestions for Authors

Di Pietro et al. have compared the pathologies, immune cells, gene upregulation, and viral and cytokine gene levels in the presence of SARS-CoV-2 coinfection with H1N1 influenza in Syrian hamsters. When given H1N1 48 hr prior to SARS-CoV-2 the Syrian hamsters were protected against weight loss and lung pathologies observed in SARS-CoV-2 single infection. Using RNA sequencing of the nasal epithelium from the H1N1, SARS-CoV-2 and dual infections the upregulation of innate and antiviral pathways was demonstrated by the time of SARS-CoV-2 inoculation in the 48 hr dual infected animals. Viral gene expression levels were measured between days 1 and 10 for the H1N1 M gene and the SARS-CoV-2 RdRp gene. Here it was unsurprising that the H1N1 M gene was expressed at similar levels on day 2 in the 48 hr dual infection animals as day 4 in the H1N1 singly infected animals. Interestingly, the SARS-CoV-2 RdRp gene was lower in the 48hr dual infection at all time points when compared with the singly infected animals. 

The authors noted that co-infection of SARS-CoV-2 with influenza is associated with significantly worse outcomes for patients. Whilst I appreciate the limitations of in vivo studies and the sensitivity of the ethics associated with these studies it might be worthwhile including some discussion on the use of other animal models such as non-human primates.

Minor issues

Figure 6 figure legend needs attention. The figure has A-D, the legend has A-C and in B is referring to another B. 

Page 14 line 517. “peaking peak” 

Page 17 line 618. Missing bracket

Author Response

Response to Reviewer 3

We thank the reviewer for their very timely review and helpful comments. We have addressed these fully.

  1. The authors noted that co-infection of SARS-CoV-2 with influenza is associated with significantly worse outcomes for patients. Whilst I appreciate the limitations of in vivo studies and the sensitivity of the ethics associated with these studies it might be worthwhile including some discussion on the use of other animal models such as non-human primates.

Response: We have included results in comparable studies in mice, ferrets and hamsters in the discussion in the original submission.  Regarding coinfection in NHPs, we can find no publications of influenza/SARS-CoV-2 coinfections, although there are many studies of single infections in a wide range of primates with pathology differing by viral strain. Small animal studies can eludicate viral interactions in the host, and extending these to NHPs may refine our immunologic understanding of mechanisms, but may not be justified in the light of the many reports of influenza/SARS-CoV-2 coinfections in humans. We respectfully point out that the variety of patient outcomes is associated with variables such as age, obesity socio-economic status and co-morbidity that are not easily modeled in animals.

  1. Figure 6 figure legend needs attention. The figure has A-D, the legend has A-C and in B is referring to another B. 

Response: The legend in the original document did have D, page 26, line 10 but this appeared to be deleted in the formatted document.  “D. Bubble graph illustrating REACTOME pathway analysis of significantly differentially expressed genes (padj>0.05) at Day 1 in H1N1<SARS-CoV-2 dual infected hamsters compared to SARS-CoV-2 singly infected hamsters…” In addition, the title had also been altered during formatting – we have corrected the legend and title. 

Page 14 line 517. “peaking peak” 

Response:  Thank you – corrected to “, with expression of both peaking at 4 dpi”

Page 17 line 618. Missing bracket

Response: my apologies, but I cannot find this error